# Dental and Dental Hygiene Students’ Knowledge and Capacity to Discriminate the Developmental Defects of Enamel: A Self-Submitted Questionnaire Survey

**DOI:** 10.3390/children9111759

**Published:** 2022-11-16

**Authors:** Maria Grazia Cagetti, Claudia Salerno, Giuliana Bontà, Anna Bisanti, Cinzia Maspero, Gianluca Martino Tartaglia, Guglielmo Campus

**Affiliations:** 1Department of Biomedical, Surgical and Dental Sciences, University of Milan, Via Beldiletto 1, 20142 Milan, Italy; 2UOC Maxillo-Facial Surgery and Dentistry Fondazione IRCCS Cà Granda, Ospedale Maggiore Policlinico, University of Milan, 20100 Milan, Italy; 3Department of Restorative, Preventive and Pediatric Dentistry, University of Bern, Freiburgstrasse 7, 3012 Bern, Switzerland; 4Department of Surgery, Microsurgery and Medicine Sciences, School of Dentistry, University of Sassari, Viale San Pietro 3/c, 07100 Sassari, Italy

**Keywords:** DDE, dental students, dental hygiene students, molar incisor hypomineralization, enamel hypomineralization, amelogenesis imperfecta

## Abstract

*Background*: A prompt and accurate diagnosis of developmental defects of enamel (DDE) is mandatory for proper treatment management. This cross-sectional survey, designed and carried out using anonymous self-administered questionnaires, aimed to assess dental and dental hygiene students’ knowledge and their capability to identify different enamel development defects. *Methods*: The questionnaire consisted of twenty-eight closed-ended questions. Two different samples of undergraduate students were selected and enrolled: a group of dental hygiene (GDH) students and a group of dental (GD) students. A multivariate logistic regression was performed by adopting the correct answers as explanatory variables to assess the difference between the two groups. *Results*: Overall, 301 completed questionnaires were analyzed: 157 from the GDH and 144 from the GD. The dental student group showed better knowledge than the GDH of enamel hypomineralization and hypoplasia (*p* = 0.03 for both). A quarter (25.25%) of the total sample correctly identified the period of development of dental fluorosis with a statistically significant difference between the groups (*p* < 0.01). Amelogenesis imperfecta (AI) was identified as a genetic disease by 64.45% of the sample, with a better performance from the GD (*p* = 0.01), while no statistical differences were found between the groups regarding molar incisor hypomineralization. Multivariate analysis showed that AI (OR = 0.40, [0.23;0.69], *p* < 0.01) and caries lesion (OR = 0.58, [0.34;0.94], *p* = 0.03) were better recognized by the GD. *Conclusions*: Disparities exist in the knowledge and management of DDE among dental and dental hygiene students in Italy; however, significant knowledge gaps were found in both groups. Education on the diagnosis and treatment of DDE during the training for dental and dental hygiene students needs to be strongly implemented.

## 1. Introduction

Developmental defects of the enamel (DDEs) are non-carious enamel conditions that comprise a heterogeneous group of defects with varying severity and etiology. These disorders occur during amelogenesis due to many potential damaging agents [1].

The prevalence of DDE is reportedly increasing worldwide, although is difficult to accurately assess their frequency as the high prevalence of dental caries can mask their presence [2]. The characteristics of the defects vary widely in terms of size, color, and shape, affecting both primary and permanent dentitions and being localized to one or more teeth. In affected teeth, the structure of the enamel differs greatly; it may consist of a quantitative defect of the enamel, i.e., hypoplasia, or a qualitative defect, i.e., hypomineralization that appears as an opacity, or a combination of the two types of defects. The degree of the defect is related to the time of amelogenesis. During the period of enamel growth, several conditions can adversely affect the enamel-forming cells [3]. If the condition acts in the secretory phase, hypoplasia occurs; if in the maturation phase, hypomineralization occurs [2]. Since enamel is unable to remodel itself, any aberration that occurs during its formation is permanently recorded on the surface. The defects’ location indicates the timing of the events that disrupted the tissue formation. A good knowledge of the chronology of tooth development can be helpful in estimating the approximate timing of the insult [4].

Defects in enamel development are clinically significant. Defects can have an affect either from an aesthetic point of view, or by causing severe hypersensitivity or impairing the masticatory function [5]. In addition, lesions located in posterior teeth are recognized as an important risk factor for caries development [6]. The etiology of DDE lesions often remains unknown, and their management is not consistently supported by evidence-based treatments [7].

Affected teeth might be discolored, and they are often sensitive and prone to fracture or wear easily (post-eruptive breakdown). Furthermore, the restoration of affected teeth is frequently very problematic as conventional dental materials do not bond effectively to the defective enamel. In addition, affected individuals may present high levels of dental anxiety, making their management challenging and stressful for the patient, as well as for the parents and clinicians [2].

Several indices have been proposed over the past 50 years to classify enamel defects, resulting in a lack of comparability between studies on enamel defects [8]. The Working Group of the FDI Commission on Oral Health, Research and Epidemiology was established in 1977. The Group recommended the use of a descriptive index called the Developmental Defects of Enamel (DDE) Index in which the type (opacity, hypoplasia, discoloration), number (single or multiple), demarcation (demarcated or diffuse), and the location of defects on the buccal and lingual surfaces of teeth are recorded [8]. Several versions of the original DDE index were later proposed [9,10,11]. The DDE index can provide valuable support to clinicians in framing defects; however, an etiologic diagnosis can only be postulated by combining the clinical aspect with a historical investigation, for example, molar incisor hypomineralization or amelogenesis imperfecta [10].

Not all DDE require treatment. If the defect does not create problems with sensitivity, occlusion, mastication, and aesthetics, simple monitoring is the most appropriate treatment choice [12]. Unless the defect is located in an aesthetic area and does not cause sensitivity, treatment can be delayed; otherwise, if the defect causes pain and progresses rapidly towards a post-eruptive breakdown and caries development, effective and timely treatment must be implemented [13]. For a proper treatment management plan, an accurate diagnosis of the type of defect is mandatory, including the accurate collection of anamnestic data and a meticulous clinical evaluation of the defect’s appearance.

During undergraduate courses in Dentistry and Dental Hygiene, students should acquire knowledge as well as several practical skills, ranging from laboratory work to diagnostic measures and a variety of therapeutic interventions. During pre-clinical classes, students are expected to acquire the knowledge needed to diagnose the different pathologies that enamel can present. In Italy, the undergraduate degree in Dentistry lasts for 6 years with five courses in which knowledge about DDE is provided with regard to their preventive and therapeutic management. The Dental Hygiene degree lasts for 3 years with about three courses in which knowledge about DDE is provided [14].

Thus, the aim of this survey was to investigate dental and dental hygiene students’ knowledge and their ability to differentiate different developmental defects of enamel. To achieve this goal, a self-submitted anonymous questionnaire was developed and distributed online.

## 2. Materials and Methods

### 2.1. Study Sample

This online-based cross-sectional survey was conducted throughout Italy with students attending the last two years of the degree in Dental Hygiene and the last three years of the degree in Dentistry. The STROBE guidelines for cross-sectional studies and the guidelines of the Declaration of Helsinki were carefully observed during the research [15,16]. The study protocol was approved by the Ethic Committee of the University of Sassari, Italy (N°AOU_SS 26) on 11 December 2020. The research was conducted at the Department of Surgery, Microsurgery and Medicine Sciences, School of Dentistry, University of Sassari.

Only students who had already attended DDE courses in accordance with their curriculum were considered eligible, therefore students attending the last two years of the degree in Dental Hygiene and the last three years of the degree in Dentistry were included. Before being enrolled, all students had to sign an informed consent form allowing the researchers to use their questionnaire data.

The minimum sample size appraised was 258 subjects, calculated with a statistical power of 90% within a significance level of 5%, considering the entire sample of student in the last two years of Dental Hygiene degrees and the last three years of the Dentistry degrees in Italy was about 5500 students [17,18].

The sample was divided according to the type of school attended: a group of dental hygiene (GDH) students and a group of students in dentistry (GD). Failure to sign informed consent, failure to submit the questionnaire and failure to answer all items were the exclusion criteria. Data were collected between March and April 2022.

### 2.2. Questionnaire

The anonymous questionnaire consisted of 28 closed-ended questions in the form of dichotomous, multiple-choice or Likert scales (Appendix A). The items investigated the basic knowledge acquired about DDE and the ability to distinguish among different developmental defects of enamel through the presentation of five clinical images (Figure 1).

Prior to the actual survey, a pre-test and the standardization of the questionnaire were performed to check its comprehensibility and quality. The questionnaire was pre-tested for comprehensibility on a small sample of 20 students (10 Dental students and 10 Dental Hygiene students) not included in the study sample. After completion, the subjects were contacted to find out whether they had experienced any difficulty in understanding the questions, assigning a comprehension score from 1 (extreme difficulty) to 5 (no difficulty) with a result of 4.6 ± 0.34.

The only demographic data required were sex, degree studied and year of study.

The first clinical image showed molars affected by severe molar incisor hypomineralization (MIH), the second image showed an entire permanent dentition affected by amelogenesis imperfecta (AI) (hypomineralized type), the third image showed upper canines and premolars affected by dental fluorosis (moderate grade following Dean classification [19]) and the last two images showed two different caries lesions (ICDAS score of 2 and 4 [20], respectively). After each image, the participant had to answer whether the defect presented a low, medium or high risk of caries and which treatment was appropriate for the type of defect depicted.

A link to the questionnaire hosted on Google Forms (Google, Mountain View, CA, USA) was sent to the students by the faculty’s secretaries to ensure the anonymity of the survey. Responses were automatically collected in Google Forms and a data sheet was generated and exported in Microsoft Excel^®^ (Microsoft, Redmond, WA, USA). The data was then cleaned, coded and the quantitative data obtained were tabulated and used for statistical analysis.

### 2.3. Statistical Analysis

For items with only one correct response, the remaining incorrect answers were merged for analysis. Discrete variables were expressed as absolute and relative frequencies (%) and compared with Pearson’s chi-squared test or Fisher’s exact test. Alpha risk was set to 5% and two-tailed tests were used. Multivariate logistic regression was performed to assess whether dental students were better at identifying different clinical images than dental hygiene students. For multivariate logistic regression analysis, the two groups were used as the dependent variable and the correct responses related to the framing of different clinical images as explanatory variables. The data were checked for multicollinearity using the Belsley–Kuh–Welsch technique. Heteroskedasticity and normality of the residuals were assessed with the White test and the Shapiro–Wilk test, respectively. A *p*-value < 0.05 was considered statistically significant. Due to the nature of the self-administered and cluster-distributed questionnaire in the different faculties, no analysis on response adherence was conducted. Statistical analysis was performed with EasyMedStat (EasyMedStat, Levallois-Perret, France) [21].

## 3. Results

The questionnaire link was opened by 323 students; 22 questionnaires were not completed and were excluded. In the end, 301 completed questionnaires were analyzed: 157 from the GDH and 144 from the GD.

Most of the questionnaires (70.43%) were filled in by women. Almost all of the sample (92.69%) claimed to have received information about DDE (91.72% in GDH and 93.75% in GD). A statistically significant difference between the two groups was found regarding how the information on DDE where obtained (*p* < 0.05). Most of the GDH (43.95 %) received information about DDE only from university courses, while the majority of the GD (55.56%) received information from both university courses and non-university activities.

Enamel hypomineralization as a qualitative defect was indicated by almost two thirds of participants (64.78%; 58.60% of GDH and 71.53% of GD, *p* = 0.03) while the percentage of correct answers rose to 69.44 % (63.69% of GDH and 75.69% of GD, *p* = 0.03) in recognizing enamel hypoplasia as a quantitative defect. Only 25.25% of the entire sample (33.76% of GDH and 15.97% of GD, *p* < 0.01) was aware that dental fluorosis is a DDE that occurs in the pre-eruptive phase, and of these, 61.84% believed that fluorosis could be easily confused with plaque demineralization. Amelogenesis imperfecta was recognized as a genetic disease by 64.45% of the entire sample (57.32% of GDH and 72.22% of GD, *p* = 0.01) and 77.41% of the total students (74.52% of GDH and 80.56% of GD) were aware that the disease develops in the pre-eruptive phase with no statistical differences among the two groups. Only 53.49 % of the whole sample (48.41% in GDH and 59.93% in GD) knew that molar incisor hypomineralization is a qualitative enamel defect and a smaller percentage (47.51%; 56.05% of GDH and 48.61% of GD) knew that it develops during the pre-eruptive phase. Both questions posed on molar incisor hypomineralization showed no statistically significant differences between the two groups. When asked how confident the students were in their ability to distinguish between DDE, only 39.53% of the sample said they were absolutely sure or fairly sure. The answers to the above questions for the two groups are presented in Table 1.

Regarding the five clinical images, only 52.81% of the entire sample correctly recognized molar incisor hypomineralization; 55.15% of the sample felt that the teeth shown had a high risk of caries and 72.43% said that fluoride or other remineralizing agents should be an appropriate treatment. Only 31.23% felt that professional whitening was an appropriate treatment to improve the aesthetic of the affected teeth (Table 2).

The students performed better (66.78%) in recognizing the second clinical picture in which teeth affected by amelogenesis imperfecta were shown. Almost all students (90.37%) believed that the affected teeth had a high caries risk and 66.11% claimed that the use of fluoride or other remineralizing agents was a desirable treatment. More than 30% of the sample would use sealants, 43.85% would use a glass-ionomer cement and 33.89% a resin-based material. Only 74.09% of the sample agreed that professional bleaching was not a suitable treatment to improve the aesthetics of the affected teeth.

Only 35.55 % of the sample correctly identified the third clinical case (dental fluorosis). A high or medium caries risk was claimed by 81.06 % of the sample; 64.12% chose the use of fluoride or other remineralizing products to treat affected teeth. Only 20.27% of the entire sample would use professional whitening to improve the aesthetics.

About one-third of the sample (32.88%) correctly identified the fourth clinical picture (white spot lesion). However, most students (75.74%) judged the affected teeth to be at high or medium caries risk, but only 61.13% stated that fluoride or other remineralizing products would be an appropriate treatment. Professional bleaching was chosen by 21.93% to improve the aesthetics.

Only 37.54 % of the sample correctly recognized the last clinical picture (caries lesion). More than half of the entire sample (63.79%) stated that fluoride or other remineralizing agents would be an appropriate treatment. More than half of the sample would use sealants, 57.48% would use a glass-ionomer cement and 55.81%, a resin-based material. Professional bleaching to improve the aesthetics was proposed by 10.63% of the sample. The results of the comparison between the two groups along with the response rates are shown in Table 2.

Multivariate logistic regression was performed to assess whether dental students were better at identifying different clinical images than dental hygiene students. Multivariate analysis (Table 3) shows that amelogenesis imperfecta (OR = 0.40, [0.23;0.69], *p* < 0.01) and caries lesion (OR = 0.58, [0.34;0.94], *p* = 0.03) were recognized more frequently by dental students than dental hygiene students, while no statistically significant differences were found for the other clinical images showing molar incisor hypomineralization, dental fluorosis and white spot lesion.

## 4. Discussion

This is the first survey to assess dental and dental hygiene students’ knowledge about DDE and their ability to correctly identify different developmental defects of enamel and confounding clinical images.

Students showed the least competence in regard to dental fluorosis and MIH, with a significant difference between dental and dental hygiene students in favor of hygiene students for fluorosis. About a quarter of the sample knew that fluorosis falls under DDE classification and is formed in the pre-eruptive phase. Regarding the concept of enamel hypomineralization and hypoplasia, dental students were more capable than hygiene students in correctly classifying these conditions. Although most of the sample recognized enamel hypomineralization as a quality defect, about half of the sample failed to recognize MIH. It is quite unexpected that the results obtained are still far from satisfactory in terms of identifying DDEs as pre-defined defect conditions [22]. Thus, these findings highlight important gaps in both groups regarding their basic knowledge about DDE. Given the high prevalence of DDE in the primary dentition (about 30%) and in the permanent dentition (from 9 to 36%) [23,24,25,26], dental professionals should be aware of these enamel defects as they are a frequent occurrence in daily clinical practice.

To avoid bias in the analysis of the comparison between the two groups, it was decided to analyze only those answers that provided a diagnostic hypothesis, thus the “don’t know” answers were excluded. A statistically significant difference was found between the two groups; the dental students performed better on the ability to distinguish between different clinical images showing MIH, amelogenesis imperfecta and caries lesions. In contrast, a statistically significant difference was found for fluorosis knowledge, with the dental hygiene students performing better. The treatment options chosen by all students also reflected a lack of knowledge about this EDD. This could be explained by the low prevalence of fluorosis in Italy, which could lead teachers to neglect this topic during courses [26].

The ability to identify severe MIH lesions was higher in the GD group compared to that described in the literature [27], where the participants were students and not professional dentists [27]. Appropriate treatments (i.e., the use of remineralizing agents and glass-ionomer sealants) were correctly selected by more hygiene students than dental students, underlying the importance of dental staff (dentists and dental hygienists) being well prepared to promote oral health through oral hygiene recommendations. Various treatment options are available and recommended, such as dietary advice, the prescription of fluoride products and other remineralizing products such as casein phosphopeptide-amorphous calcium phosphate [28]. The recommendation on the use of GIC versus resin-based materials is dependent on the chemical bonding of GIC and the capacity to release fluoride [29,30].

Amelogenesis imperfecta was recognized more frequently by dental students, although hygiene students were more likely to consider individuals with AI at high risk of caries. No statistically significant differences were found on the type of treatment proposed, with the exception of remineralizing agents, which were strongly recommended by the hygiene students. Surprisingly, a small percentage of students in both groups suggested professional bleaching for this DDE. This cosmetic treatment should not be performed on heavily hypomineralized enamel as it can severely worsen sensitivity and contribute to tissue demineralization [31].

Most of the sample confused dental fluorosis with plaque demineralization. Although the two conditions share some traits, fluorosis is a symmetrical defect, potentially affecting all tooth surfaces, unlike plaque demineralization which is mainly localized to the sites of plaque accumulation [32]. Although hygiene students were more able to recognize this type of DDE, they were more likely to incorrectly consider the subject at high risk of caries. Again, remineralizing agents were more strongly recommended by dental hygiene students.

Disappointing results were found for plaque demineralization lesions (white spots), which were correctly identified by only one-third of the sample, and almost half of the hygiene students rated the caries risk as high, while most dental students rated the caries risk as medium with a statistically significant difference. This finding also reflects the deficiencies of most students with regard to caries classification indices and treatment. The last image, “a cavitated occlusal carious lesion” was most frequently recognized by dental students, but overall, only one-third of the total sample made a correct assessment. Regarding the type of treatment recommended, most of the sample suggested potentially suitable strategies, with the exception of bleaching, which was considered useful by only a small percentage of students.

Because early diagnosis is essential to establish effective treatment and to avoid the risk of rapid lesion progression, a multivariate analysis was performed to assess whether either group had a higher chance of avoiding misdiagnosis [13]. With regard to amelogenesis imperfecta and caries lesion, dental students demonstrated a significantly lower OR than hygiene students in the multivariate analysis. This result suggests that dental students, despite studying for more years and taking more courses on the topic, did not always show significantly better knowledge about DDE than dental hygiene students. This finding underlines the importance of the role that dental hygienists, as competent and qualified health professionals, play within a dental team,, as previously reported [33].

One of the main strengths of this study, to the best of the authors’ knowledge, is the novelty of the subject of this survey, which assessed dental and dental hygiene students’ knowledge of EDDs and their ability to distinguish between different defects. The results fill a gap in the literature, and provide an indication of which areas of student preparation need to be improved. One limitation of the study is the imbalance regarding sex, as the sample included a high prevalence of females. This reflects the fact that dental hygiene is a female-dominated profession, and also the difference between the two sexes in their compliance to this kind of investigation [34]. A further limitation could be the different size and type of electronic devices that were used to fill in the questionnaire, which could affect the captioning of the details of the images shown and thereby their recognition.

## 5. Conclusions

Disparities exist in the knowledge and management of DDE among dental and dental hygiene students in Italy, with significant knowledge gaps found in both groups. Education on the diagnosis and treatment of DDE during the training of students in both courses need to be strongly implemented. The results of this survey highlight the urgent need to improve students’ basic knowledge of DDEs and their management. Since the students surveyed showed particular difficulties in diagnosing the clinical conditions in the images they were shown, it follows that the teaching of DDEs would benefit from training based on clinical images that facilitate the recognition of different conditions so as to improve the ability to make differential diagnoses between developmental and acquired enamel defects.

## Figures and Tables

**Figure 1 children-09-01759-f001:**
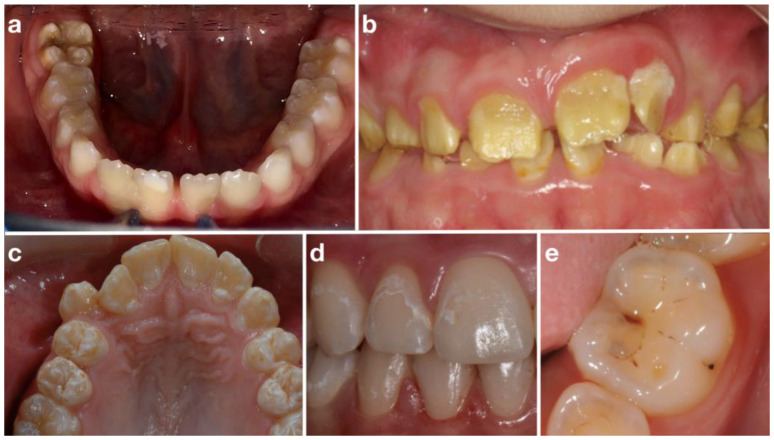
Clinical images showed in the questionnaire: (**a**), molar incisor hypomineralization; (**b**), amelogenesis imperfecta; (**c**), dental fluorosis; (**d**), white spot lesion; (**e**), caries lesion.

**Table 1 children-09-01759-t001:** DDE knowledge among GDH and GD (expressed as number and percentage). The difference between groups was analyzed by Pearson’s chi-square test.

Items	GDH N = 157	GD N = 144	*p*-Value
**Enamel hypomineralization is a …… defect:**			
Qualitative	92 (58.60%)	103 (71.53%)	**0.03**
Quantitative/Both/I don’t know	65 (41.4%)	41 (28.47%)	
**Enamel hypoplasia is a ……. defect:**			
Quantitative	100 (63.69%)	109 (75.69%)	**0.03**
Qualitative/Both/I don’t know	57 (36.31%)	35 (24.31%)	
**When does Dental Fluorosis develop?**			
In the pre-eruptive phase	53 (33.76%)	23 (15.97%)	**<0.01**
In the post-eruptive phase/any age/I don’t know	104 (66.24%)	121 (84.03%)	
**Amelogenesis Imperfecta is a condition caused by:**			
Genetic factors	90 (57.32%)	104 (72.22%)	**0.01**
Systemic factors/Multifactorial/viral or bacterial infection/Don’t know	67 (42.68%)	40 (27.78%)	
**When does Amelogenesis Imperfecta develop?**			
In the pre-eruptive phase	117 (74.52%)	116 (80.56%)	**0.26**
In the post-eruptive phase/any age/I don’t know	40 (25.48%)	28 (19.44%)	
**Molar Incisor Hypomineralization is a …… defect:**			
Qualitative	76 (48.41%)	85 (59.03%)	0.08
Quantitative/Both/Don’t know	81 (51.59%)	59 (40.97%)	
**When does Molar Incisor Hypomineralization develop?**			
In the pre-eruptive phase	88 (56.05%)	70 (48.61%)	0.24
In the post-eruptive phase/any age/I don’t know	69 (43.95%)	74 (51.39%)	

**Table 2 children-09-01759-t002:** Comparison of GDH and GD answers in regard to the clinical images (expressed as number and percentage). Differences between the groups were analyzed by Pearson’s chi-square test or Fisher’s exact test.

Items	GDH	GD	*p*-Value
**Picture 1: Molar Incisor Hypomineralization**	N = 137	N = 121	
Possible choices			
Molar Incisor Hypomineralization	83 (60.58%)	91 (75.21%)	**0.04**
Amelogenesis Imperfecta	24 (17.52%)	9 (7.44%)	
Plaque demineralization	17 (12.41%)	13 (10.74%)	
Dental Fluorosis	13 (9.49%)	8 (6.61%)	
I don’t know	20/157 (12.74%)	23/144 (15.97%)	
The caries risk in this situation is generally:			
High	98 (71.53%)	48 (39.67%)	**<0.01**
Medium	38 (27.74%)	68 (56.20%)	
Low	1 (0.73%)	5 (4.13%)	
Which of the following treatments would you recommend?			
Remineralizing products and/or fluoride-based varnish or gel	119 (86.86%)	84 (69.42%)	**0.01**
Glass-ionomer sealants	63(45.99%)	52 (42.98%)	0.72
Resin-based sealants	51 (37.23%)	49 (40.50%)	0.68
Professional bleaching	28 (20.44%)	15 (12.40%)	0.12
**Picture 2: Amelogenesis Imperfecta**	N = 131	N = 135	
Possible choices			
Amelogenesis Imperfecta	89 (67.94%)	112 (82.96%)	**0.02**
Plaque demineralization	25 (19.08%)	10 (7.41%)	
Molar Incisor Hypomineralization	12 (9.16%)	10 (67.41%)	
Dental Fluorosis	5 (3.82%)	3 (2.22%)	
I don’t know	26/157 (16.56%)	9/144(6.25%)	
The caries risk in this situation is generally:			
High	125 (95.42%)	117(86.67%)	**0.02**
Medium	6 (4.58%)	16 (11.85%)	
Low	0 (0.00%)	2 (1.48%)	
Which of the following treatments would you recommend?			
Remineralizing products and/or fluoride-based varnish or gel	100 (76.34%)	86 (63.7%)	**0.03**
Glass-ionomer sealants	66 (50.38%)	55 (40.74%)	0.16
Resin-based sealants	53 (40.46%)	41(30.37%)	0.11
Professional bleaching	14 (10.69%)	15 (11.11%)	>0.99
**Picture 3: Dental Fluorosis**	N = 146	N = 133	
Possible choices			
Dental Fluorosis	61 (41.78%)	46 (34.59%)	**0.01**
Plaque demineralization	46 (31.51%)	66 (49.62%)	
Molar Incisor Hypomineralization	33 (22.60%)	18 (13.53%)	
Amelogenesis Imperfecta	6 (4.11%)	3 (2.26%)	
I don’t know	11/157 (7.01%)	11/144 (7.64%)	
The caries risk in this situation is generally:			
High	53 (36.30%)	25 (18.80%)	**<0.01**
Medium	71 (48.63%)	83 (62.41%)	
Low	22 (15.07%)	25 (18.80%)	
Which of the following treatments would you recommend?			
Remineralizing products and/or fluoride-based varnish or gel	104 (71.23%)	78 (58.65%)	**0.04**
Glass-ionomer sealants	74 (50.68%)	70 (52.63%)	0.84
Resin-based sealants	88 (60.27%)	72 (54.14%)	0.36
Professional bleaching	31 (21.23%)	24 (18.05%)	0.60
**Picture 4: White spot lesion**	N = 133	N = 112	
Possible choices			
Plaque demineralization	54 (40.60%)	45 (40.18%)	0.22
Dental Fluorosis	61 (45.86%)	60 (53.57%)	
Amelogenesis Imperfecta	11 (8.27%)	3 (2.68%)	
Molar Incisor Hypomineralization	7 (5.26%)	4 (3.57%)	
I don’t know	24/157 (15.29%)	32/144 (22.22%)	
The caries risk in this situation is generally:			
High	62 (46.62%)	17 (15.18%)	**<0.01**
Medium	53 (39.85%)	62 (55.36%)	
Low	18 (13.53%)	33 (29.46%)	
Which of the following treatment would you recommend?			
Remineralizing products and/or fluoride-based varnish or gel	91 (68.42%)	73 (65.18%)	0.69
Glass-ionomer sealants	57 (42.86%)	56 (50.00%)	0.32
Resin-based sealants	58 (43.61%)	68 (60.71%)	**0.01**
Professional bleaching	35 (26.32%)	21 (18.75%)	0.21
**Picture 5: Caries lesion**	N = 121	N = 104	
Possible choices			
Plaque demineralization	48 (39.67%)	65 (62.50%)	**<0.01**
Molar Incisor Hypomineralization	67 (55.37%)	37 (35.58%)	
Dental Fluorosis	4 (3.31%)	1 (0.96%)	
Amelogenesis Imperfecta	2 (1.65%)	1 (0.96%)	
I don’t know	36/157 (22.93%)	40/104 (27.78%)	
Which of the following treatments would you recommend?			
Remineralizing products and/or fluoride-based varnish or gel	91 (75.21%)	76 (73.08%)	0.83
Glass-ionomer sealants	86 (71.07%)	66 (63.46%)	0.28
Resin-based sealants	71 (58.68%)	71 (68.27%)	0.18
Professional bleaching	9 (7.44%)	15 (14.42%)	0.14
For each clinical pictures’ comparison analysis, the answer “I don’t know” was excluded			

**Table 3 children-09-01759-t003:** Multivariate logistic regression analysis between GD and GDH on their recognition of clinical images. The correct answer was chosen as a reference.

Ref: *Group of students in Dentistry (GD)*		
Clinical Pictures	Odds Ratio	*p*-Value
Molar Incisor Hypomineralization	0.93 [0.56;1.54]	0.77
2. Amelogenesis Imperfecta	0.40 [0.23;0.69]	<0.01
3. Dental Fluorosis	1.38 [0.81;2.35]	0.23
4. White spot lesion	1.30 [0.75;2.24]	0.35
5. Caries lesion	0.58 [0.34;0.94]	0.03

## Data Availability

Not applicable.

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
