# Peer review of "Dental and Dental Hygiene Students’ Knowledge and Capacity to Discriminate the Developmental Defects of Enamel: A Self-Submitted Questionnaire Survey"

_children, 2022, doi:10.3390/children9111759_

Round 1

Reviewer 1 Report

Manuscript of considerable interest that increasingly increases the training method of students in dental hygiene and dentistry

Well described abstract

Sufficient keywords

Introduction; add recent research published by Prof Scribante's research group, published in Children, on the various factors that induce MIH. https://doi.org/10.3390/children8060432

Materials and methods, how was the sample size calculated?

Results well described

Discussion, to add as future objectives the correct analysis of all the teachings and disciplines concerning the degree course of dental hygiene and dentistry, increasing the skills of the students as the research group of Prof Lupi has already published on Healthcare mdpi https: // doi.org/10.3390/healthcare10010115

Well described conclusions.

Bibliography, add required references

Reviewer 2 Report

Thank you for allowing me to review this scientific manuscript. The manuscript is interesting. However, it has a certain number of shortcomings, which I list below.

1. Abstract, authors stated: "Dental student group showed better knowledge of enamel hypomineralisation and hypoplasia (p<0.05); a quarter (25.25%) of the sample correctly identified the aetiology and development of dental fluorosis with a statistically significant difference between the groups (p<0.01)." - separate into two sentences, and state the p values.

2. Abstract/conclusion - the first sentence is incomprehensible... BUT???

3. Materials and methods - indicate at which department and study the research was conducted.

4. Materials and methods - size is usually calculated with a 95% confidence interval, a 5% margin of error, and a 50% population proportion, meaning there should be 385 respondents out of 5500 students. This is a failed study; it can be a pilot or find a reference and refer to why the 90% confidence interval was taken.

5. Materials and methods - which are included and which exclusion criteria? What year of study were the students?

6. Results, the authors stated: "The questionnaire link was sent to 323 students; 22 questionnaires were not completed and were excluded. In the end, 301 completed questionnaires were analyzed: 157 from GDH and 144 from GD." - How many of the 323 sent were GDH, and how many GD students? Why only 323 students? How many are GDH students, and how many GD students are there? The total size must be 385 or calculated separately for GDH and GD students.

7. Discussion - Please indicate the strength of the study.

8. Conclusion - too general; it should be more specific.

9. References - please check; for example book number 2 is not well written.

Round 2

Reviewer 1 Report

The manuscript has been correctly revised, it can be published

Reviewer 2 Report

Thanks to the authors for accepting some requested modifications, however the book references are still not written according to the instructions. I don't see why this is such a problem.

I can ignore the insufficient sample of 258 respondents, which is small compared to the examined population of 5500, so next time, let the authors try harder. And let them list the number of respondents in the limiting factors of the study.

Since the changes are not made in Track Changes, it is difficult to follow it.